# Guided Path Sampling: Steering Diffusion Models Back on Track with Principled Path Guidance

## Abstract

Iterative refinement methods based on a denoising-inversion cycle are powerful tools for enhancing the quality and control of diffusion models. However, their effectiveness is critically limited when combined with standard Classifier-Free Guidance (CFG). We identify a fundamental limitation: CFG's extrapolative nature systematically pushes the sampling path off the data manifold, causing the approximation error to diverge and undermining the refinement process. To address this, we propose Guided Path Sampling (GPS), a new paradigm for iterative refinement. GPS replaces unstable extrapolation with a principled, manifold-constrained interpolation, ensuring the sampling path remains on the data manifold. We theoretically prove that this correction transforms the error series from unbounded amplification to strictly bounded, guaranteeing stability. Furthermore, we devise an optimal scheduling strategy that dynamically adjusts guidance strength, aligning semantic injection with the model's natural coarse-to-fine generation process. Extensive experiments on modern backbones like SDXL and Hunyuan-DiT show that GPS outperforms existing methods in both perceptual quality and complex prompt adherence. For instance, GPS achieves a superior ImageReward of 0.79 and HPS v2 of 0.2995 on SDXL, while improving overall semantic alignment accuracy on GenEval to 57.45%. Our work establishes that path stability is a prerequisite for effective iterative refinement, and GPS provides a robust framework to achieve it.

## Keywords

Text-to-image, Diffusion Models, Off-Manifold

## 1 INTRODUCTION

Diffusion models have emerged as the dominant paradigm for high-fidelity signal generation [2, 16, 20, 22, 27], with their control largely specified through sampling techniques [17, 28]. The popular of these, which we term Z-sampling[1], leverages Classifier-Free Guidance (CFG)[11] to effectively steer generation. They found that the guidance gap between denoising and inversion could accumulate semantic information and the process of repeatedly applying an inversion-denoising cycle serves to maximize the integration of semantic information. While powerful, we find that Z-sampling suffers from a crucial "off-manifold" limitation [13]. Our analysis, supported by mathematical proof, shows that CFG's core "extrapolation" strategy causes guidance errors to systematically diverge, pushing the denoised estimate away from the true data manifold. This issue presents a primary bottleneck for achieving complex, detailed control [4, 9].

The consequences of this off-manifold problem are especially severe for advanced iterative refinement methods built upon the cycle. During Z-sampling, each CFG-guided denoising step introduces an error by pushing the estimate off-manifold. Because this estimate is no longer on the valid data distribution, the subsequent

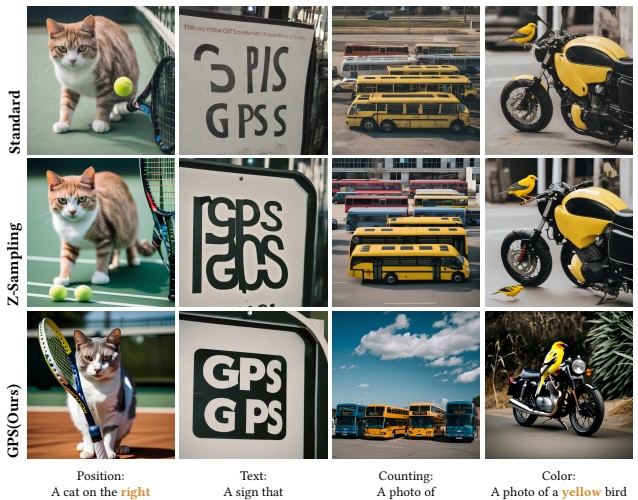

Position: A cat on the **right** of a tennis racket.   Text: A sign that says **'GPS'**   Counting: A photo of **four** buses   Color: A photo of a **yellow** bird and **black** motorcycle

**Figure 1: Visual comparison of GPS with baselines on challenging prompts. GPS demonstrates superior performance across tasks involving spatial positioning, text rendering, object counting, and attribute binding. Unlike Standard and Z-Sampling methods which suffer from artifacts or semantic misalignment, GPS maintains high fidelity and prompt adherence.**

inversion step is inherently inaccurate [19]. This error is then fed back into the next iteration, where it is compounded by further CFG extrapolation . The continuous introduction and amplification of this error throughout the cycle not only causes visual artifacts and oversaturation but critically disrupts the invertibility of samplers like DDIM [28], ultimately rendering these powerful refinement techniques ineffective.

To counteract CFG's error divergence, one could theoretically use high-order solvers [12], but this is computationally inefficient. We therefore propose a fundamentally different and efficient paradigm, guided by the geometric imperative to remain on the data manifold: **G**uided **P**ath **S**ampling (GPS). At its core, GPS re-engineers the denoising-inversion cycle with a manifold-constrained interpolation mechanism [6], replacing unstable extrapolation. This elegantly resolves the cumulative error problem. Drawing further inspiration from the coarse-to-fine nature of diffusion [5], GPS dynamically schedules the guidance strength, which drastically reduces artifacts and enhances semantic fidelity.

Our contributions are twofold. **Theoretically**, we prove that GPS stabilizes iterative refinement by constraining the approximation error within a strictly bounded convex hull, thereby preventing divergence. We also show that a monotonically increasing cosine schedule for guidance strength better simulates cognitive refinement. **Experimentally**, GPS demonstrates marked superiority over

standard and Z-sampling methods across multiple benchmarks. On SDXL [20], it achieves a leading ImageReward of 0.79 and HPS v2 of 0.2995, significantly surpassing the Z-sampling baseline. This superiority extends to the transformer-based Hunyuan-DiT [15], where GPS sets a new benchmark with an ImageReward of 0.97. Furthermore, on GenEval [7], GPS improves overall prompt alignment accuracy to 57.45%, with notable gains in complex compositional tasks. These results validate GPS as a more robust and effective solution for high-fidelity, controllable generation [26].

## 2 RELATED WORKS

**Off-Manifold problems in diffusion models**   Recent studies have shown that CFG can introduce systematic *off-manifold* errors during the denoising process. Specifically, the linear extrapolation step in CFG pushes intermediate estimates away from the true data manifold $\mathcal{M}$ [3]. Consequently, iterative refinement schemes that rely on a *denoising–inversion* cycle accumulate these deviations, yielding visible artifacts and violating the invertibility guarantees of deterministic samplers such as DDIM.

While interpreting the sampling process as solving an ordinary or stochastic differential equation (ODE/SDE) [29] can partially mitigate the drift, these solvers incur a significant computational overhead that is impractical for interactive editing or real-time applications. To address this limitation, manifold-preserving techniques have been proposed: (i) *geometric projection* methods that explicitly project back onto $\mathcal{M}$; (ii) *energy-guided* samplers that penalize off-manifold deviations with learned energy functions [18, 25]; and (iii) *shortcut* algorithms that re-parameterize the sampling path to remain within a learned latent subspace [8]. Despite these advances, existing approaches either require auxiliary networks or rely on costly optimization loops.

## 3 METHODOLOGY

In this section, we first analyze the origin of systematic error in iterative samplers, then introduce a manifold constraint to resolve it, and finally derive our method **GPS**, based on theoretical analysis.

### 3.1 Preliminaries & Definitions

To formally ground our analysis, we first establish the key preliminaries and definitions and a core assumption about the data manifold.

#### 3.1.1 Denoising Diffusion Implicit Models.
DDIM introduce a deterministic sampling process that is also invertible. This process is defined by a pair of single-step mappings: a denoising operation $\mathcal{D}$ and its exact inverse $\mathcal{I}$.

The denoising map $\mathcal{D}$ computes a less noisy data sample $\mathbf{x}_{t-1}$ from $\mathbf{x}_t$ as:

$$\mathbf{x}_{t-1} = \mathcal{D}(\mathbf{x}_t) = \sqrt{\bar{\alpha}_{t-1}}\left(\frac{\mathbf{x}_t - \sqrt{1-\bar{\alpha}_t}\,\mathbf{x}_t^{\omega}}{\sqrt{\bar{\alpha}_t}}\right) + \sqrt{1-\bar{\alpha}_{t-1}}\,\mathbf{x}_t^{\omega}$$

Conversely, the inversion map $\mathcal{I}$ reconstructs the noisier sample $\mathbf{x}_t$ from $\mathbf{x}_{t-1}$:

$$\tilde{\mathbf{x}}_t = \mathcal{I}(\mathbf{x}_{t-1}) = \frac{\sqrt{\bar{\alpha}_t}}{\sqrt{\bar{\alpha}_{t-1}}}\mathbf{x}_{t-1} + \left(\sqrt{1-\bar{\alpha}_t} - \frac{\sqrt{\bar{\alpha}_t(1-\bar{\alpha}_{t-1})}}{\sqrt{\bar{\alpha}_{t-1}}}\right)\mathbf{x}_{t-1}^{\omega}$$

Here, $\mathbf{x}_t$ denotes the latent state at timestep $t$, while $\mathbf{x}_t^{\omega}$ represents the noise estimate predicted by the U-Net backbone [24] under the classifier-free guidance scale $\omega$. The term $\bar{\alpha}_t$, defined as $\prod_{i=1}^{t}(1-\beta_i)$, characterizes the cumulative noise schedule, where $\beta_i$ governs the variance of the Gaussian noise injected at each forward step.

#### 3.1.2 Zigzag Sampling.
For $t = T, \ldots, T - K$, alternate:

$$\textbf{Zig:} \quad \mathbf{x}_{t-1} = \mathcal{D}(\mathbf{x}_t \mid c, \omega_h),$$
$$\textbf{Zag:} \quad \tilde{\mathbf{x}}_t = \mathcal{I}(\mathbf{x}_{t-1} \mid c, \omega_l).$$

Z-sampling refines the sampling path by repeatedly alternating between a "Zig" step with a high guidance scale $\omega_h$ and a "Zag" step with a low guidance scale $\omega_l$. This iterative process is applied for the initial timesteps before switching to a standard denoising procedure to obtain the final clean image.

*Definition 3.1 (Semantic Information Gain).* We define the *semantic information gain term* $\boldsymbol{\tau}_1(t)$ as the difference between the noise estimates:

$$\boldsymbol{\tau}_1(t) = \mathbf{x}_t - \tilde{\mathbf{x}}_t. \tag{1}$$

This gain is scheduled to be proportional to the difference in guidance scales, denoted by $\delta_\omega$, such that:

$$\boldsymbol{\tau}_1(t) \propto \delta_\omega \quad \text{where} \quad \delta_\omega = \omega_1 - \omega_2. \tag{2}$$

*Definition 3.2 (Approximation Error and its Decomposition).* The single-step approximation error $\boldsymbol{\tau}_2(t)$ in Z-Sampling is defined and decomposed as:

$$\boldsymbol{\tau}_2(t) = \tilde{\mathbf{x}}_t - \tilde{\mathbf{x}}_{t-1} = \boldsymbol{\tau}_{\text{manifold}}(t) + \boldsymbol{\tau}_{\text{local}}(t). \tag{3}$$

The two components are defined as:

- **Local Discretization Error** ($\boldsymbol{\tau}_{\text{local}}$): The ideal error between two on-manifold points:

$$\boldsymbol{\tau}_{\text{local}}(t) := \tilde{\mathbf{x}}_t^{on} - \mathbf{x}_{t-1}^{on} \tag{4}$$

- **Systematic Manifold-Offset Error** ($\boldsymbol{\tau}_{\text{manifold}}$): The error induced by the guidance mechanism:

$$\boldsymbol{\tau}_{\text{manifold}}(t) := (\tilde{\mathbf{x}}_t - \tilde{\mathbf{x}}_t^{on}) - (\tilde{\mathbf{x}}_{t-1} - \mathbf{x}_{t-1}^{on}) \tag{5}$$

*Definition 3.3 (Guidance Mechanisms).* We distinguish between two guidance mechanisms. **Off-Manifold Guidance** uses extrapolation ($\omega > 1$), pushing the estimate $\mathbf{x}_t^{\omega}$ off the on-manifold path. In contrast, our **Manifold-Constrained Guidance** uses interpolation ($\lambda \in [0, 1]$), ensuring the estimate $\mathbf{x}_t^{\lambda}$ remains on the path. The respective estimates are:

$$\mathbf{x}_t^{\omega} = (1 - \omega)\mathbf{x}_t^{\phi} + \omega\mathbf{x}_t^{c} \tag{6}$$
$$\mathbf{x}_t^{\lambda} = (1 - \lambda)\mathbf{x}_t^{\phi} + \lambda\mathbf{x}_t^{c} \tag{7}$$

### 3.2 Guided Path Sampling

The cumulative effect of the approximation error $\boldsymbol{\tau}_2(t)$ directly determines the final image quality. We argue that standard CFG's use of *Off-Manifold Guidance* is the root cause of performance degradation in iterative samplers like Z-Sampling. This continuous off-manifold guidance introduces an ineliminable systematic error, ultimately causing the cumulative error series to diverge.

**Algorithm 1** GPS

1: **Input:** Text prompt $c$, Denoising operation $\mathfrak{D}$, Inversion operation $\mathfrak{I}$, denoising guidance $\lambda_1 \in [0, 1]$, inversion guidance scheduling function $\lambda_2(t)$, total inference steps $T$, self-reflection steps $K$.
2: **Output:** Clean image $\mathbf{x}_0$.
3: Sample Gaussian noise $\mathbf{x}_T \sim \mathcal{N}(0, \mathbf{I})$.
4: **for** $t = T$ to $1$ **do**
5:     **if** $t > T - K$ **then**
6:         $\mathbf{x}'_{t-1} \leftarrow \mathfrak{D}(\mathbf{x}_t, c, \lambda_1)$
7:         $\lambda_{2,t} \leftarrow \lambda_2(t)$
8:         $\tilde{\mathbf{x}}_t \leftarrow \mathfrak{I}(\mathbf{x}'_{t-1}, c, \lambda_{2,t})$
9:         $\mathbf{x}_t \leftarrow \tilde{\mathbf{x}}_t$
10:     **end if**
11:     $\mathbf{x}_{t-1} \leftarrow \mathfrak{D}(\mathbf{x}_t, c, \lambda_1)$
12: **end for**
13: **return** $\mathbf{x}_0$

THEOREM 3.4 (ERROR DIVERGENCE OF Z-SAMPLING). *Assume:*

- *The noise prediction function is twice continuously differentiable near the data manifold.*

*Then for Z-Sampling with CFG scale $\omega > 1$, the cumulative inversion error diverges:*

$$\sum_{t=1}^{T} \|\boldsymbol{\tau}_2(t)\| \to \infty \quad as \; T \to \infty.$$

PROOF. Let $\mathbf{d}(\mathbf{x}_t) := \mathbf{x}_t^c - \mathbf{x}_t^\phi$. At step $t$, the off-manifold perturbation magnitude is:

$$\|\boldsymbol{\delta}_{t-1}\| \approx \sqrt{\bar{\alpha}_{t-1}} \, (\omega - 1) \, \|\mathbf{d}(\mathbf{x}_t)\|.$$

Assuming the manifold has non-vanishing curvature represented by the tensor $\mathcal{H}$, the second-order manifold error satisfies:

$$\|\boldsymbol{\tau}_{\text{manifold}}(t)\| \approx \tfrac{1}{2}\|\mathcal{H}(\xi)[\boldsymbol{\delta}_{t-1}, \boldsymbol{\delta}_{t-1}]\| \geq \kappa\|\boldsymbol{\delta}_{t-1}\|^2,$$

where $\kappa > 0$ relates to the manifold curvature. Thus,

$$\|\boldsymbol{\tau}_{\text{manifold}}(t)\| \gtrsim \bar{\alpha}_{t-1}(\omega - 1)^2\|\mathbf{d}(\mathbf{x}_t)\|^2.$$

Since the guidance term $\mathbf{d}(\mathbf{x}_t)$ represents a semantic direction independent of the step size $\Delta t$, this error term is $O(1)$ with respect to $T$. Consequently, summing this non-vanishing error over $T$ steps leads to divergence:

$$\sum_{t=1}^{T} \|\boldsymbol{\tau}_2(t)\| \geq \sum_{t=1}^{T} c = \Omega(T) \to \infty.$$

$$\mathfrak{D}(\mathbf{x}_t) = \sqrt{\bar{\alpha}_{t-1}}\left(\frac{\mathbf{x}_t - \sqrt{1 - \bar{\alpha}_t}\, \mathbf{x}_t^\lambda}{\sqrt{\bar{\alpha}_t}}\right) + \sqrt{1 - \bar{\alpha}_{t-1}}\, \mathbf{x}_t^\phi \quad (8)$$

$$\mathfrak{I}(\mathbf{x}_{t-1}) = \sqrt{\bar{\alpha}_t}\left(\frac{\mathbf{x}_{t-1} - \sqrt{1 - \bar{\alpha}_{t-1}}\, \mathbf{x}_{t-1}^\lambda}{\sqrt{\bar{\alpha}_{t-1}}}\right) + \sqrt{1 - \bar{\alpha}_t}\, \mathbf{x}_{t-1}^\phi \quad (9)$$

To resolve the error divergence issue, we propose **Guided Path Sampling (GPS)**. Its core idea is to adopt the *Manifold-Constrained Guidance* defined in Section 3.1, which fundamentally eliminates the systematic manifold offset error. We then replace part of the guided

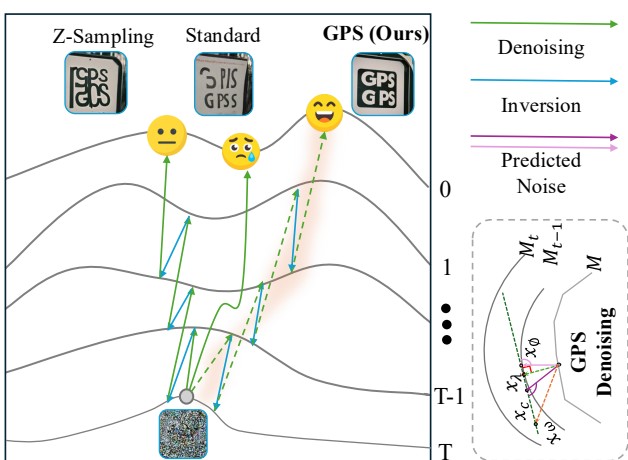

**Figure 2: Schematic illustration of GPS.** Unlike standard methods that extrapolate off the manifold (red dotted line), GPS employs a manifold-constrained interpolation (green solid line) during the zigzag cycle, ensuring the sampling trajectory remains stable and errors remain bounded.

noise with unconditional noise, as defined in equations $\mathfrak{D}(\mathbf{x}_t)$ and $\mathfrak{I}(\mathbf{x}_{t-1})$. The complete procedure is detailed in Algorithm 1. The stability of this approach is guaranteed by our first core theorem.

THEOREM 3.5 (ERROR BOUNDEDNESS OF GPS). *Let the noise predictions be bounded. For GPS employing manifold-constrained guidance (interpolation), the cumulative approximation error $\sum_{t=1}^{T} \|\boldsymbol{\tau}_2(t)\|$ is **strictly bounded**, ensuring sampling stability.*

PROOF. Let $\mathbf{x}_t^\phi$ and $\mathbf{x}_t^c$ denote the unconditional and conditional noise predictions, respectively, assumed to be bounded in magnitude by a constant $M$.

Recall that standard CFG (Eq. 6) employs extrapolation with $\omega > 1$, which amplifies the deviation:

$$\|\mathbf{x}_t^\omega\| = \|(1 - \omega)\mathbf{x}_t^\phi + \omega\mathbf{x}_t^c\| = \|\mathbf{x}_t^\phi + \omega(\mathbf{x}_t^c - \mathbf{x}_t^\phi)\|.$$

As $\omega$ increases, the norm $\|\mathbf{x}_t^\omega\|$ grows linearly with $\omega$, potentially becoming unbounded and pushing the trajectory off-manifold.

In contrast, GPS (Eq. 7) employs interpolation with $\lambda \in [0, 1]$:

$$\mathbf{x}_t^\lambda = (1 - \lambda)\mathbf{x}_t^\phi + \lambda\mathbf{x}_t^c.$$

By the Triangle Inequality, the magnitude of the guided noise is strictly bounded by the convex hull of the component predictions:

$$\|\mathbf{x}_t^\lambda\| \leq (1 - \lambda)\|\mathbf{x}_t^\phi\| + \lambda\|\mathbf{x}_t^c\| \leq \max(\|\mathbf{x}_t^\phi\|, \|\mathbf{x}_t^c\|) \leq M.$$

Consequently, the manifold offset error $\boldsymbol{\tau}_{\text{manifold}}(t)$ does not diverge. The total cumulative error is dominated by the local discretization error, which is bounded for the finite time horizon $T$:

$$\sum_{t=1}^{T} \|\boldsymbol{\tau}_2(t)\| \leq \sum_{t=1}^{T} (C \cdot \Delta t) = C \cdot T\Delta t = O(1).$$

Thus, the error series remains bounded, guaranteeing algorithmic stability.

# 4 Experiments

## 4.1 Experimental Setup

**Datasets.** We evaluate our model on two complementary benchmarks. For assessing human-perceived **aesthetic quality**, we use the first 100 prompts from Pick-a-Pic [14]. For quantitatively measuring **compositional accuracy** (e.g., object count and position), we use GenEval. This dual evaluation provides a holistic view of our model's capabilities.

**Metrics** We evaluate text-image alignment using CLIP Score [10], and complement it with HPS v2 [30] and ImageReward (IR) [31], two learned metrics trained on extensive human preference judgments to capture subjective quality.

**Diffusion Models** We employ different diffusion models as the generation backbone in our experiments. For SD2.1 [23], SDXL [21], and Hunyuan-DiT [15], we perform 50 denoising steps. We set $\omega = 5.5$, $\lambda_1 = 0.5$ and use $\lambda_{2,t}$ that increases from 0.1 to 0.3 using a cosine function in SDXL/SD2.1, and $\omega = 6.0$ in Hunyuan-DiT, aligning with the default recommended values. Finally, the zigzag operation is executed along the entire path ($K = T - 1$).

## 4.2 Main Results

Table 1: Comparative results on the Pick-a-Pic benchmark.

|  | Method | CLIP ↑ | HPS v2 ↑ | IR ↑ |
|---|---|---|---|---|
| SDXL | Standard | 0.710 | 0.2899 | 0.64 |
|  | Z-Sampling | 0.719 | 0.2980 | 0.75 |
|  | GPS | **0.723** | **0.2995** | **0.79** |
| SD-2.1 | Standard | 0.681 | 0.2541 | -0.54 |
|  | Z-Sampling | 0.696 | 0.2686 | -0.25 |
|  | GPS | **0.702** | **0.2709** | **-0.18** |
| Hunyuan-DiT | Standard | 0.712 | 0.2915 | 0.92 |
|  | Z-Sampling | 0.724 | 0.3012 | 0.94 |
|  | GPS | **0.730** | **0.3056** | **0.97** |

Table 2: Comparative results on GenEval with SDXL.

| Metric | Standard | Z-Sampling | GPS (ours) |
|---|---|---|---|
| Single Obj. | 97.50% | **100.00%** | **100.00%** |
| Two Obj. | 69.70% | 74.75% | **76.77%** |
| Count. | 33.75% | 46.25% | **48.75%** |
| Colors | 86.71% | **87.23%** | 86.17% |
| Pos. | 10.00% | 10.00% | **11.00%** |
| Color Attr. | 18.00% | **24.00%** | 22.00% |
| **Overall** | 52.52% | 57.04% | **57.45%** |

As presented in Table 1 and 2, GPS consistently outperforms both Standard sampling and Z-Sampling methods across varying benchmarks and model architectures.

On the Pick-a-Pic benchmark (Table 1), GPS demonstrates superior performance in text-image alignment (CLIP) and human preference metrics (HPS v2, IR). Notably, on the SDXL backbone, GPS

achieves an ImageReward of **0.79** and HPS v2 of **0.2995**, surpassing the strong Z-Sampling baseline. This superiority extends to different architectures, including the older SD-2.1 and the Transformer-based Hunyuan-DiT, where GPS sets a new state-of-the-art with an ImageReward of **0.97**. These results indicate that maintaining the manifold structure effectively enhances both semantic fidelity and aesthetic quality.

Table 2 further details the fine-grained semantic capabilities on SDXL using the GenEval benchmark. GPS achieves the highest **Overall** score of **57.45%**. Crucially, significant gains are observed in complex compositional tasks, such as **Counting** (48.75% vs. 46.25% for Z-Sampling) and **Two Object** generation (76.77% vs. 74.75%). This suggests that our stable iterative refinement better accumulates semantic details and spatial structures for complex prompts, validating the effectiveness of our manifold constraints.

## 4.3 Ablation Study

Table 3: Ablation of the inversion scheduler $\lambda_{2,t}$ on SDXL.

| Scheduler | CLIP ↑ | HPS v2 ↑ | IR ↑ |
|---|---|---|---|
| Constant (0.1) | 0.711 | 0.2983 | 0.72 |
| Constant (0.3) | 0.714 | 0.2986 | 0.73 |
| Sigmoid (0.1→0.3) | 0.719 | 0.2993 | 0.74 |
| Linear (0.1→0.3) | 0.721 | 0.2994 | 0.75 |
| **Cos (0.1→0.3)** | **0.723** | **0.2995** | **0.79** |
| Cos (0.3→0.1) | 0.717 | 0.2992 | 0.74 |
| Cos (0.1→0.3→0.1) | 0.710 | 0.2983 | 0.72 |

Our ablation study on the inversion guidance scheduler (Table 3) provides strong empirical backing for our theory. We evaluated various strategies for scheduling the guidance scale $\lambda_{2,t}$, ranging from fixed `Constant` values to dynamic schedules like `Cos (0.1→0.3)`. The results across all metrics (CLIP, HPS v2, and IR) reveal two key findings.

First, dynamic scheduling consistently surpasses constant schedules, with the `Cos (0.1→0.3)` strategy achieving the highest scores (e.g., **0.79** IR and **0.2995** HPS v2). Second, and most importantly, the results directly validate our proposed **"coarse-to-fine" refinement principle**: monotonically increasing schedules for $\lambda_{2,t}$ yield the best performance. This confirms that gradually strengthening the manifold constraint is the optimal strategy. Conversely, schedules that violate this principle by decreasing the scale (`Cos (0.3→0.1)`) or being non-monotonic (`Cos (0.1→0.3→0.1)`) fail to maintain this benefit, leading to a noticeable degradation in both semantic alignment and perceptual quality.

# 5 CONCLUSION

We identified that standard extrapolative CFG pushes sampling paths off-manifold, causing error divergence. We introduced GPS, which replaces extrapolation with manifold-constrained interpolation, transforming the divergent error of methods like Z-Sampling into a provably convergent process. Furthermore, we proposed an optimal, monotonically increasing guidance schedule to align semantic injection with the model's coarse-to-fine generation . Our

experiments show GPS significantly improves perceptual quality and semantic alignment. **The key takeaway is that path stability is a prerequisite for effective iterative refinement.** Future work will focus on extending GPS to stochastic samplers and exploring learned scheduling functions.

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
