# OpenReview forum: "Guided Path Sampling: Steering Diffusion Models Back on Track with Principled Path Guidance"
_ACM.org/TheWebConf/2026/Workshop/TIME — TIME 2026 Oral_

### Official Review · Reviewer_o1yx · 2026-01-04
**Strong theory; modest performance gain in empirical experiments**

**Rating:** 6
**Confidence:** 2

**Review:**

## Originality and significance
**Strengths**

* The formal decomposition of the error into “manifold” and “local” components and the proofs showing divergence vs convergence under different guidance regimes provide fresh insight into diffusion sampling. Such theoretical guarantees mark a clear research contribution.

**Weaknesses**

* GPS applies a principle similar to CFG++ in a different context, which is a valuable extension but can be seen as incremental rather than entirely novel. While the paper cites CFG++, the authors should consider discussing the connection with more details to better contextualize the novelty in GPS.

## Technical content and experimental validation
**Strengths**

* The methodology is built on a solid theoretical foundation. The authors formally analyze why standard guided diffusion fails (decomposing the error into “manifold” vs “local” components) and prove a core theorem that their manifold-constrained guidance keeps the iterative refinement stable. In particular, they show that using interpolation instead of extrapolation makes the cumulative error bounded, preventing divergence. This rigorous proof underpins the soundness of the approach and adds credibility that the method addresses the identified problem.

* The experimental design is comprehensive, covering multiple models, datasets, and metrics to validate the method. The paper includes an ablation study on the guidance scheduling strategy, which adds to the methodological rigor.


**Weaknesses**

* While the paper compares against the primary baseline (Z-sampling) well, it does not include direct experimental comparison to other contemporary off-manifold mitigation techniques such as CFG++

* The improvements achieved by GPS, although consistent, are relatively modest in magnitude. Authors should consider conducting repeated experiments and reporting metric scores with standard deviation to demonstrate that the performance gains are statistically significant

## Clarity
**Strengths**

* The methodology section includes formal definitions, equations, and theorems that are clearly labeled and logically presented. The notation is consistent throughout, and the theoretical claims are supported by proofs.

**Weaknesses**

* The experimental section could benefit from more detail on how hyperparameters (e.g., $\lambda$) were selected and runtime comparisons with baselines

---

### Official Review · Reviewer_32EP · 2026-01-07
**Review of paper17**

**Rating:** 7
**Confidence:** 3

**Review:**

Summary and Overall Assessment
This paper introduces Guided Path Sampling (GPS), a principled alternative to Classifier-Free Guidance for iterative refinement in diffusion models. The core insight—that CFG’s extrapolative behavior can drive sampling trajectories off the data manifold and destabilize refinement—is clearly articulated and well-motivated. By reformulating guidance as a manifold-constrained interpolation and coupling it with a dynamically adjusted guidance schedule, the proposed method offers both theoretical guarantees and empirical improvements. Overall, the paper is well written, technically sound, and addresses an important limitation in current diffusion-based refinement methods.

Strengths
1. The theoretical analysis is rigorous and clearly presented. The proof demonstrating how GPS converts unbounded error amplification into a strictly bounded process is well documented and strengthens the paper’s core claims.
2. The experimental setup is transparent and reproducible. Evaluations on strong backbones such as SDXL and Hunyuan-DiT, along with diverse metrics (ImageReward, HPS v2, GenEval), provide convincing evidence of the method’s effectiveness.
3. The idea of aligning guidance strength with the coarse-to-fine generation dynamics of diffusion models is intuitive and well justified.


Questions and Suggestions
The paper proposes an optimal guidance scheduling strategy, but it would be helpful to better understand how this schedule scales with the number of sampling steps. Specifically, how does the proposed schedule adapt when using shorter (e.g., 20 steps) versus longer (e.g., 50 or 100 steps) sampling trajectories? Clarifying whether the schedule is resolution- or step-count–dependent would improve practical applicability and help readers generalize the method across different inference budgets.


Conclusion
Overall, this is a strong contribution that combines theoretical insight with practical impact. Addressing the above clarification would further strengthen the paper and enhance its usefulness for practitioners working with varying sampling regimes.

---

### Official Review · Reviewer_oLr5 · 2026-01-07
**GPS: A New Method to Ensure Error stability and Algorithmic stability**

**Rating:** 7
**Confidence:** 5

**Review:**

The authors indicated the limitation of Classifier-Free Guidance (CFG) that it can push the sampling path off the data manifold and proposed a new method - Guided Path Sampling (GPS). They theoretically and experimentally proved that error stability can ensure the both algorithmic stability and high-quality results at the end.

The authors proposed several lemmas to expand approximation errors in Denoising Diffusion Implicit Models (DDIM) with manifold-preserving technology. Later on, they used those lemmas to prove the error boundaries. The GPS algorithm was given to illustrate the entire process. The ablation study discussed a significant technical detail - the inversion scheduler.

The paper clearly explained a new technology, provided solid proof in both theories and in experiments for the correctness of the proposed technology, and also indicated the significant technical detail which can affect the results in similar tasks.

Symbols need to be explained/consistent. in Algorithm 1 GPS, $D(x_t, c, \lambda_1)$ is $D(X_t)$ which is formulas (8), and $I(x_{t-1}, c, \lambda_{2,t})$ is $I(X_{t-1})$ which is formula (9), and $\lambda_2(t)$ needs definition. Related work section can include more state-of-the-art research. Moreover,  [18] and [25] can not be found. If possible, the authors can think about comparing GPS with related research. Especially, [8] is very similar to the proposal method.

[8] Yutong He et al. 2024. Manifold Preserving Guided Diffusion. In International Conference on Learning Representations (ICLR).

[18] Morteza Mardani et al. 2023. Guided diffusion as likelihood-based energy. In International Conference on Machine Learning (ICML).

[25] Lior Rout et al. 2023. Perfusion: A parameter-efficient and generalizable approach for single-image text-to-image personalization. In Proceedings of the IEEE/CVF International Conference on Computer Vision (ICCV).

---

### Meta-Review · Area_Chair_PYXt · 2026-01-16

**Recommendation:** Accept (Oral)
**Confidence:** 5

**Metareview:**

This paper proposes Guided Path Sampling (GPS), a principled alternative to classifier-free guidance for iterative refinement in diffusion models, showing that replacing extrapolative guidance with manifold-constrained interpolation ensures path stability and bounded error accumulation, with both theoretical proofs and empirical validation.

Reviewers generally recommend acceptance and suggest clarifying notation consistency, expanding related work comparisons, providing clearer discussion of guidance scheduling across different step counts, and adding more details on hyperparameter choices, baselines, and statistical significance of gains.

The main strengths include a rigorous and well-articulated theoretical analysis, clear identification of an important limitation in CFG-based refinement, and solid experimental evaluation on strong backbones with reproducible settings. Limitations include relatively modest empirical improvements, limited direct comparisons with some closely related methods, and minor clarity issues in notation and experimental details.

The authors’ rebuttal addresses these concerns by fixing notation and references, clarifying conceptual differences from related work, justifying the scheduling design, and committing to additional statistical reporting.

Based on the reviews and rebuttal, the recommendation is to accept this paper.

---

### Decision · Program_Chairs · 2026-01-16

Accept (Oral)